# Morphometric Study of the Eyeball of the Loggerhead Turtle (*Caretta caretta*) Using Computed Tomography (CT)

**DOI:** 10.3390/ani13061016

**Published:** 2023-03-10

**Authors:** Marcos Fumero-Hernández, Mario Encinoso Quintana, Ana S. Ramírez, Inmaculada Morales Fariña, Pascual Calabuig, José Raduan Jaber

**Affiliations:** 1Veterinary Clinical Hospital, Faculty of Veterinary Medicine, University of Las Palmas de Gran Canaria, 35413 Las Palmas, Spain; 2Department of Pathology and Food Technology, Faculty of Veterinary Medicine, Universidad de Las Palmas de Gran Canaria, 35413 Las Palmas, Spain; 3Tafira Wildlife Rehabilitation Center (Cabildo de Gran Canaria), 35017 Las Palmas, Spain; 4Department of Morphology, Faculty of Veterinary Medicine, University of Las Palmas de Gran Canaria, 35413 Las Palmas, Spain

**Keywords:** computed tomography, morphometry, eyeball, sclerotic ring, anatomy, loggerhead turtle

## Abstract

**Simple Summary:**

There are few publications describing the use of advanced diagnostic imaging techniques in turtle ophthalmology. However, adequate knowledge of the anatomy of the eye and its associated structures, given their adaptations to be able to develop in the aquatic environment and also in the terrestrial one, could be an important tool to deepen the ophthalmological anatomy, lifestyle, and pathology of these animals.

**Abstract:**

The short bibliography referring to the anatomy and pathology of the eyeball of turtles poses a challenge for veterinarians and conservationists given the increasing presence of this type of turtle in veterinary and wildlife centres. Although they nest on land, these animals spend a large part of their lives in the ocean, which entails a series of eye adaptations such as well-developed nictitating membranes, palpebral scales, highly sensitive corneas, or sclerotic rings to protect the eye. In our study, we performed a morphometric analysis of the loggerhead turtle (*Caretta caretta*) eyeball and its internal structures using advanced imaging techniques such as computed tomography (CT). To the best of the authors’ knowledge, there have been no studies published that describe the CT intraocular measurements of presumed normal loggerhead turtle eyes.

## 1. Introduction

The loggerhead turtle (*Caretta caretta*) is a species of sea turtle that shows a wide distribution, mainly in all the world’s temperate and subtropical regions of the Atlantic, Pacific, and Indian oceans, as well as the Mediterranean Sea [1]. In the wintertime, they also carry out large migrations to tropical and subtropical waters to deposit their eggs on sandy beaches, where the hatchlings move to the open sea to feed. Loggerhead sea turtles are primarily carnivorous, but they can also eat other types of food, making them omnivorous. The loggerhead head is relatively large. It has a roughly triangular shape, wide posteriorly, and the snout tapers anteriorly to the orbits. Moreover, the head presents deep parietal notches as happens in other sea turtle species such as the olive ridley or the Kemp’s ridley turtles [2,3]. Compared to other sea turtles, loggerheads have a relatively long secondary palate that separates or partially separates food and air passages. The considerable development of their jaws, which are robust and shaped in a wide V, serve as an effective tool for dismantling their pray, allowing them to eat hard-shelled prey and other invertebrates, such as sponges, jellyfish, cephalopods, shrimp, insects, and sea urchins, including different fish species [2]. There are slight variations in the diet of each life stage, but loggerhead sea turtles are generalists throughout life [3]. The loggerhead sea turtle is the world’s largest hard-shelled turtle. Adult individuals of this species have an approximate weight range of 80 to 200 kg, averaging around 135 kg [2,3], and their shells can exceed a meter in length. The head and carapace range from a yellow-orange to a reddish-brown colour that can be covered with barnacles and algae, whereas the plastron is typically pale yellow. Sexual dimorphism of the loggerhead sea turtle can be observed in adults. In the juvenile stage, male specimens have increasing levels of testosterone as they approach maturity, triggering wider carapaces, longer curved claws on each forelimb [3,4], and longer, more robust tails [5,6], whereas females produce oestrogen and small amounts of testosterone, but externally just grow larger [4,7].

The loggerhead turtle is currently on the International Union for Conservation of Nature (IUCN) list of threatened species, which is considered a vulnerable species since it is in decline [8]. Among the multiple causes of decline that have included this animal on the list, we report those associated with human activity and disturbances (72%), entanglement in plastic or fishing nets (50.81%), and the ingestion of hooks (11.88%) [9]. Different studies have highlighted sense of sight as an important element for interaction with the environment and maintaining their survival [10]. Some reports have described that artificial lighting near beaches may confuse emerging hatchlings, causing them to move away from beaches and into hazardous urban areas [4,11]. Therefore, sea turtle conservation is an important concern around the world; conservation efforts include an increase in knowledge of the anatomy and physiology of stranded animals, in part because the behaviour of sea turtles has an anatomic basis [2,4]. Hence, veterinarians, biologists, and conservationists need to know the turtle’s ocular structures and their functioning that could affect the greater or lesser incidence of strandings.

Turtles present a series of ophthalmological peculiarities, some common with other groups of vertebrates such as birds. This is the case of the sclerotic rings, made up of a variable number of imbricated ossicles and which, together with the scleral cartilages, form the so-called ocular or scleral skeleton [12,13]. Its function has been scarcely described, but it is believed that it protects the eye against mechanical pressure in those species with diving capacities, support to preserve the shape, and the accommodation mechanism, allowing for the insertion of the ciliary muscle at the edge of the cornea [13,14,15,16]. Despite the significance of the sclerotic ring and the applicability of imaging modalities for anatomical evaluations in live animals, most of the studies on the biometric characteristics of the ocular skeleton concerning the eyeball have been conducted in fossils such as Caypullisaurus [17], or non-passerine bird species [18]. Only a few imaging morphometric studies have been carried out in alive animals [19,20] and, interestingly, just a description has been performed on turtles [21]. Nevertheless, specific studies concerning the ocular morphometry of these last animals are essential since they may provide information on the activity patterns of the animals, such as their diurnal or nocturnal behaviour [13,17]. Therefore, this study aimed to measure the eyeball and its structures in the loggerhead turtle (*Caretta caretta)* using non-invasive examinations such as computed tomography. 

## 2. Materials and Methods

### 2.1. Animals

To perform this study, we used 10 loggerhead turtles from the Tafira Wildlife Rehabilitation Center (Gran Canaria, Canary Islands, Spain), which weighed between 19 and 30 kg and were classified as young and subadult turtles since age is not easily quantifiable in these animals. As physical examination did not provide sufficient information to allow for a characterisation of the exact cause of stranding; the rehabilitation centre requested modern methods of non-invasive diagnostic imaging to evaluate their appendicular skeleton and metabolic bone diseases and rule out foreign bodies and lesions of the internal organs. Therefore, the turtles were enrolled in this study with the prior consent of the person in charge of the rehabilitation centre. Before performing the CT study, we completed an ophthalmologic examination of all turtles and we did not observe signs of disease in any of these animals. 

### 2.2. CT Technique

CT studies of the skull of these individuals were performed with the animals immobilised in plastic containers. These containers, provided by the wildlife center, were the place where they carried out different medical procedures. Therefore, these animals were used to spending time in containers. In addition, the room temperature (ranging between 16 and 20 °C) can produce lethargy, which is helpful when performing the imaging study without sedation. Sequential slices were obtained using a 16-slice helical CT scanner (Toshiba Astelion, Canon Medical System^®^, Tokyo, Japan). The animals were positioned symmetrically in the prone position on the stretcher, with craniocaudal entry, and a standard clinical protocol was used (120 kVp, 80 mA, 512 × 512 acquisition matrix, 1809 × 858 field of view, pitch of 0.94, and a gantry rotation of 1.5 s), to acquire images of a 0.6 mm thickness. Different CT images were obtained in dorsal, transverse, and sagittal planes with bone and soft tissue windows. All these images were uploaded to an image viewer (OsiriX MD, Apple, Cupertino, CA, USA) to perform data manipulation and measurements of the loggerhead turtle eyeball.

### 2.3. Measurements

We measured the carapace longitudinal length (CCL), including the midline between the cranial and caudal portions of this structure. In addition, the different measurements of both eyes (*n =* 20) were analysed by two observers using oblique sagittal, transverse, and dorsal CT images of ten skulls, with a soft tissue attenuation window. The parameters and measures taken are described below:(A)Transverse plane respect to the eyeball-Lens diameter, understood as the maximum distance between the lateral and medial edges of the lens (equatorial diameter) (Figure 1A).-Internal diameter of the sclerotic ring, or maximum distance between the inner lateromedial edges of the ring, close to the cornea (Figure 1A).-External diameter of the sclerotic ring or maximum distance between the outer lateromedial edges of the ring, close to the sclera (Figure 1A).(B)Transverse plane concerning the turtle’s body-Dorsal and ventral arch length: the thickness of the upper and lower regions of the scleral ring, with the corneal and scleral margins (Figure 2B).-Width of the dorsal and ventral arches of the scleral ring, that is, of the upper and lower parts of the lateral portion of the ring (Figure 2A).-Height of the eyeball or distance between the most dorsal and the most ventral portion of the eye (Figure 3B).-Attenuation of the sclerotic ring and lens, taken in the dorsal area and vitreous humour expressed in Hounsfield Units (HU) (Figure 4).(C)Dorsal plane-Width of the eyeball, which is the lateromedial length of the globe (Figure 3A).-Length of the eyeball or distance between the most rostral and caudal portion of this structure (Figure 3A).

### 2.4. Statistical Analysis

We performed the statistical analysis with commercially available software (SPSS 19, Statistical Package for the Social Sciences, Chicago, IL, USA). The mean, median, range, and standard deviation (sd) were calculated for each measurement. The Mann–Whitney U test was used to compare the measurements between the right and left eyes, and the dorsal and ventral arch. To compare differences between the right and left eyes, a two-way analysis of variance (ANOVA) test was used. A linear regression analysis was performed assessing the correlation of all variables to the carapace length, as well as to analyse the quantitative variables. The statistical significance was set at *p* < 0.05.

## 3. Results

The dorsal and transverse CT images obtained in this study were helpful to identify and measure the structures of the eyeball (Figure 1, Figure 2, Figure 3 and Figure 4). From these images, we obtained the different measurements corresponding to the right and left eyes (Table 1).

On the CT image, the eyeball appeared nearly spherical, with a soft tissue attenuating rim in all loggerhead turtles, differing in its sagittal, transverse, and vertical diameters. This border probably represents part of the sclera. Other eyeball structures that were distinguishable were the lens, aqueous and vitreous chambers, and the sclerotic rings (Figure 1). This last chamber was identified as a fluid-attenuating region that accounted subjectively for most of the eyeball volume. In addition, the lens appeared as a spherical hyperattenuated structure suspended in the eyeball and contacting with the vitreous body. The scleral ring, embedded in the rostral portion of the ocular sclera, showed a circular, continuous, and hyperattenuated appearance.

Table 1 shows the measurements made in the right and left eye and in the different planes of all the animals studied. A total of 10 loggerhead turtles were studied, with all 20 eyes measured. The mean carapace length was 41.53 cm (range= 25–71.7), the median was 39.25 cm, and the standard deviation was 13.11 cm.

The mean lens diameter was 4.33 mm for all eyes (range = 2.6–5.6 mm). The mean internal diameter of the sclerotic ring was 10.32 mm (range = 6.9–14.7), while the mean external diameter of the sclerotic ring was 16.96 mm (range = 13–22).

The means arch heights were 6.82 mm (range = 4.44–10.1) and 4.96 (range = 3.2–7.51) for the dorsal and ventral measurements, respectively. The difference between the dorsal and ventral arch height was statistically significant (*p* = 0.0006). At the same time, the means of the dorsal and ventral arch width were 2.81 mm (range = 1.89–6.71) and 2.57 mm (range = 1.8–5.56), respectively, with no significant differences.

Last, the eyeball presented a mean height of 22.23 mm (range = 15.9–27), a mean width of 15.77 mm (range = 12–19.2), and a mean length of 22.83 mm (range = 16–27.9).

Concerning attenuation, the three circles (yellow, green, and red) indicate the areas used to describe the mean attenuation of different structures of the eyeball. Therefore, the sclerotic ring (yellow circle) was hyperattenuated (mean 353.85 HU: range = 203–500), and the lens (green circle) showed median attenuation (mean 98.85 HU; range = 81–116). Nonetheless, the vitreous chamber (red circle) was hyperattenuated (mean 17.80 HU; range = 7–28) with the adjacent structures (Figure 4).

The Mann–Whitney U test revealed no significant differences when the measurements from the right and left eyes were analysed. Additionally, no statistical differences between the right and left eyes were found when all the variables measured were analysed using a two-way ANOVA. When linear regression analysis was conducted to evaluate the correlation between the measurements and the carapace length, statistical significance was found with the lens diameter (R^2^ = 0.40; *p* = 0.00290), the internal diameter of the sclerotic ring (R^2^ = 0.67; *p* = 0.00001), the external diameter of the sclerotic ring (R^2^ = 0.71; *p* = <0.00001), ventral arch height (R^2^ = 0.62; *p* = 0.00004), eyeball height (R^2^ = 0.36; *p* = 0.00557), and eyeball length (R^2^ = 0.61; *p* = 0.00005).

## 4. Discussion

Modern diagnostic imaging techniques such as computed tomography and magnetic resonance imaging have revolutionised clinical diagnosis in reptile medicine since they are valuable tools that demonstrate appreciable advantages over conventional imaging techniques. Therefore, they make it possible to obtain views of body sections from various tomographic planes, providing images with an adequate anatomic resolution without the superimposition of the tissues, a high contrast between different structures, and excellent tissue differentiation [22,23,24]. Thus, advanced imaging technologies have improved the quality of anatomical imaging and diagnosis, enabling an exceptional evaluation of different anatomical regions and a better detection of different diseases such as metabolic bone disease, skull fractures, abscesses, and neoplasia [25]. To the best of the authors’ knowledge, this is the first study reporting measurements on loggerhead turtle healthy eyes from CT scans. Hence, the different CT images of the loggerhead turtle eyeball have proven useful in this study since they provided the imaging of orbital soft tissues and ocular adnexa with a high spatial and moderate contrast resolution. We should highlight that the evaluation of anatomic structures within the heads of loggerhead sea turtles and the examination of soft tissues is arduous because of the turtle’s anatomic difficulty [26]. Non-invasive imaging exams have already been performed to describe some structures of the eye in reptiles, such as the Komodo dragon [23,24], the green iguana, the common tegu, and the bearded dragon [25], and different species of turtles inhabiting marine, freshwater, and terrestrial environments such as the green turtle, the black-bellied slider, the loggerhead turtle, and the red-footed tortoise [21,26]. Different studies have used contrast media to enhance the densities of both humours [19,27]. However, we did not use a radiocontrast since the sclerotic ring and associated ocular structures, such as the lens or the vitreous chamber, could be adequately visualised. Therefore, we identified most of the eyeball structures already seen in other studies performed in sea and terrestrial turtles [21,26] and domestic mammals such as dogs [19] and cats [27]. 

The anatomic knowledge of the eyeball of the loggerhead turtle is helpful for the ophthalmology, lifestyle, and visual ability of marine animals already described in other reports [11,12,21,28]. Our CT evaluation revealed that eyeball size and sclerotic ring width were correlated. This positive correlation between these two parameters could be associated with their level of visual acuity. Nonetheless, further studies with a larger number of animals should be done to confirm this statement. Moreover, the width and length of the dorsal arches of the sclerotic rings in all turtles were greater than those of the ventral ones. Similar findings were observed in other marine turtle species such as *Chelonia mydas*, where the sclerotic ring presented a close form, and was wider in the dorsal region than in its ventral aspect [21]. As stated in other reports, sea turtles usually dive and spend most of their lives in the marine environment. Therefore, this greater thickness of the dorsal arches of the ring could be related to the influence of water pressure on the eyes of these animals during diving [21]. Nevertheless, studies with this scope are needed to directly assess the influence of water pressure on the thickness of the ossicle rings.

Some authors have proposed scleral ossification as a possible growth marker. Nonetheless, it is important to highlight that other studies reported that scleral ossicles would not be an alternative for other sea turtle species since other sea turtles such as leatherbacks show some advantage by reaching a large size quicker, which is perhaps related to the thermoregulatory capacity that allows them to exploit a foraging niche not available to other sea turtle species [21,29]. 

Concerning attenuation variation, we recorded different values for the scleral ring, lens, and vitreous humour. Although bone densities have been reported for reptiles, such as turtles, snakes, and lizards, there are still no attenuation reference values for the eyeball structures of sea turtles [21]. Changes in some of these densities could be considered an important reason to perform a further ocular examination to determine the presence of haemorrhage, inflammation, degeneration, and neoplasia [19]. We also highlight that we did not observe any mineral attenuation structure that we could identify as scleral cartilage, unlike what has been reported by other authors in different species [21,30,31]. However, in agreement with other researchers [13,21], we did not appreciate the junction of this ring with the skull. 

In this study, we present a relatively low number of specimens of different sizes and weights. This limited number in our study was due to these animals being free-ranging turtles and due to this fact, it is not easy to obtain a large number of animals as happens in domestic mammals. Therefore, we only access them when they appear stranded on our beaches. 

In conclusion, the CT images obtained in this study provided an adequate interpretation of the anatomy of the eyeball and the sclerotic ring in live loggerhead turtles. Our reference values included eyeball and lens presumed normal diameters, as well as different measurements of the scleral ring, which could be related to the vision capability and habits of these animals. However, further studies are necessary since a limited number of animals of different sizes and weights in this study prevents us from attributing this finding to other sea turtles. Moreover, as another limitation, it is important to consider the inherent examiner error related to the manual contouring of computed tomography images.

## Figures and Tables

**Figure 1 animals-13-01016-f001:**
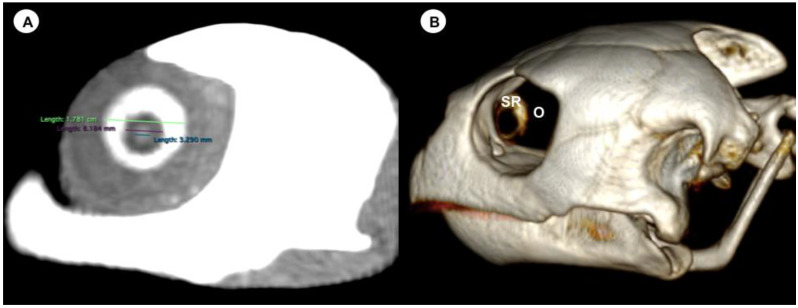
(**A**) Transverse multiplanar reconstruction (MPR) related to the eyeball of the *C. caretta* with measurements of the lens and the internal and external diameters of the sclerotic ring. (**B**) Volume rendering image of the *C. caretta* skull with the sclerotic ring (SR) and orbit (O).

**Figure 2 animals-13-01016-f002:**
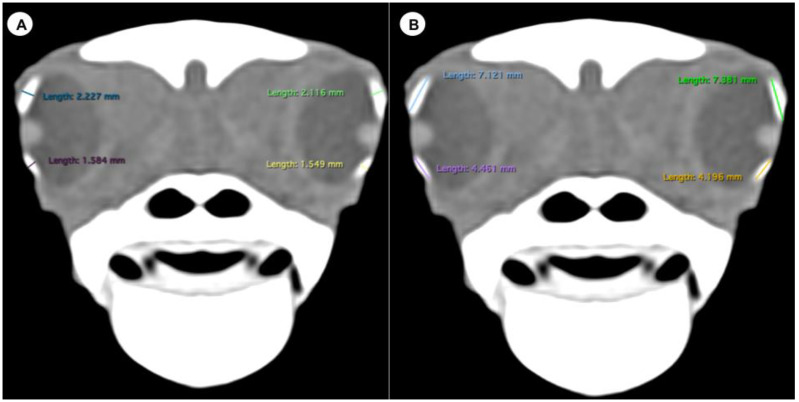
Transverse CT image in soft tissue window of the *C. caretta* head with the width (**A**) and height (**B**) of the dorsal and ventral arches of the sclerotic ring.

**Figure 3 animals-13-01016-f003:**
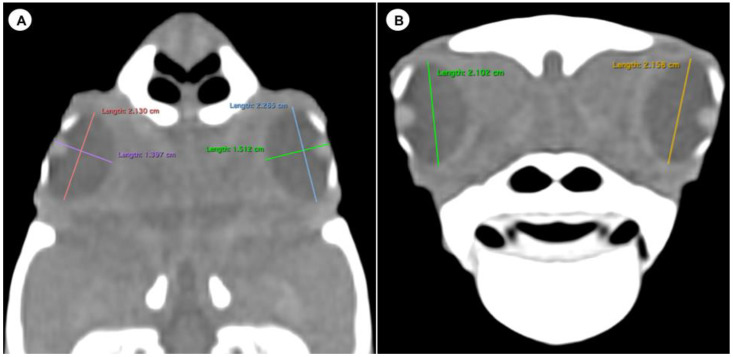
Dorsal multiplanar reconstruction (MPR) and transverse CT images in soft tissue windows of the head with measurements of the length, width (**A**), and height (**B**) of the eyeball of the *C. caretta* head.

**Figure 4 animals-13-01016-f004:**
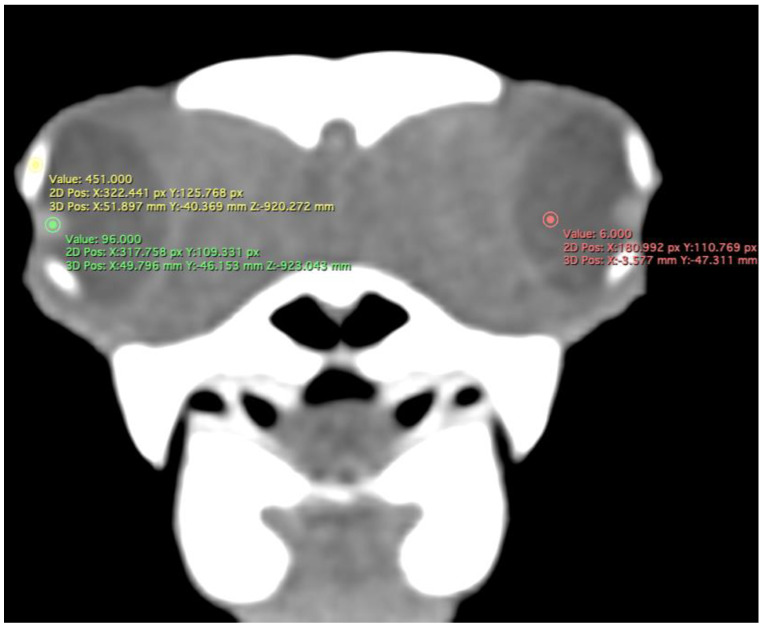
Transverse CT image in soft tissue window of the *C. caretta* head. Regions of interest (ROI) indicating the areas used to measure mean attenuation in Hounsfield Units (HU) of the lens (green circle), sclerotic ring (yellow circle), and vitreous humour (red circle).

**Table 1 animals-13-01016-t001:** Measurements of the right and left eye of the loggerhead turtle.

	Right Eye				Left Eye				Both Eyes					
	Mean	Median	Range	SD	Mean	Median	Range	SD	Mean	Median	Range	SD	r^2^	*p*
Lens diameter (mm)	4.26	4.34	2.8–5.11	0.70	4.40	4.29	2.6–5.36	0.88	4.33	4.29	2.6–5.6	0.78	0.40	**0.00290**
Internal diameter of the sclerotic ring (mm)	10.18	10.40	6.9–13.8	1.90	10.45	10.20	7.4–14.7	2.45	10.32	10.35	6.9–14.7	2.14	0.67	**0.00001**
External diameter of the sclerotic ring (mm)	16.63	16.50	13–22	2.52	17.29	16.35	13.4–22	2.97	16.96	16.35	13–22	2.70	0.71	**<0.00001**
Lens attenuation (HU)	99.10	99.50	81–116	10.13	98.60	97.50	88–108	6.57	98.85	98.00	81–116	8.31	0.11	0.15480
Vitreous humor attenuation (HU)	18.10	17.50	7–28	5.49	17.50	18.50	9–23	4.20	17.80	18.00	7–28	4.76	0.00	0.80290
Sclerotic ring attenuation (dorsal arch) (HU)	368.00	360.50	292–500	60.23	339.70	359.50	203–417	65.31	353.85	360.50	203–500	62.84	0.05	0.32200
Dorsal arch height (mm)	6.88	6.69	4.6–10.0	1.67	6.76	6.81	4.44–10	1.80	6.82	6.81	4.44–10.1	1.69	0.65	**0.00002**
Ventral arch height (mm)	4.95	4.67	3.7–7.12	1.09	4.97	4.69	3.2–7.51	1.35	4.96	4.69	3.2–7.51	1.19	0.62	**0.00004**
Dorsal arch width (mm)	2.84	2.21	2–6.04	1.59	2.79	2.66	1.89–5.5	1.50	2.81	2.36	1.89–6.71	1.23	0.43	**0.00303**
Ventral arch width (mm)	2.81	2.20	1.98–6.5	1.71	2.65	2.29	1.8–5.48	1.52	2.57	2.15	1.8–5.59	1.08	0.32	**0.01687**
Eyeball height (mm)	22.28	23.05	17–25	2.44	22.17	23.35	15.9–27	3.79	22.23	23.10	15.9–27	0.31	0.36	**0.00557**
Eyeball width (mm)	15.68	15.80	12–19.10	1.88	15.86	15.75	12–19.2	2.27	15.77	15.75	12–19.2	2.00	0.31	**0.01112**
Eyeball length (mm)	22.95	22.80	16–27.9	3.37	22.71	22.40	18–27.1	2.83	22.83	22.40	16–27.9	3.00	0.61	**0.00005**

## Data Availability

The data supporting reported results can be found at “accedacris.ulpgc.es”.

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
