# Peer review of "Morphometric Study of the Eyeball of the Loggerhead Turtle (Caretta caretta) Using Computed Tomography (CT)"

_animals, 2023, doi:10.3390/ani13061016_

Round 1
Reviewer 1 Report
The Authors describe an interesting study on loggerheads eyes, investigating the dimensions of the globe and the structures attached to the eye. Indeed, there is no information in the scientific literature about the morphometry of the eyeball. The study represents a good start, but the very reduced number of turtles and the variability of size make impossible to establish standard measures. Perhaps the statistical relationship between eye size and turtle size would have been more interesting, if the number and class age could be larger.
Specific comments:
Line 36: 2 words attached
Line 42: the diet is not related to the environment
Line 44: error
Line 47: no references declare this
Line 49: testosterone is influenced by stress
Line 51: in the red list they were moved to vulnerable
Line 52: 2 words attached
Line 59: no scientific references allow this sentence
Line 66: 2 words attached
Line 69: a space is missed
Line 77: with 10 animals any standardization is not scientific
Line 82: replace with 10
Line84-85: The animals are young and subadults, the age is not so easily quantifiable
Lines 87 -92: I understand that healthy animals were enrolled in this study, with no signs of eye disease so it would be sufficient to say they were animals without any disease.
Line 86: later referred to what?
Line 95: In my experience, in these animals anaesthesia is required to perform CT scans in order to prevent movement during the scans. How these animals were immobilized in the plastic container without anesthesia and without stress?
Line 107: in sea turtle biometric measurements CCL or SCL are used
Line 142: take off a space
Line 143: 2 words attached
Line 160: replace with tests
Line 181-182: the sentence is not clear, it needs revision
Line 212. The sentence is incorrect, please rewrite it
Line 232-233 it is not correct to write “normal” loggerhead turtle. Maybe you mean healthy eyes?
Line 244: the sentence is uncorrect: most of
Line 246-251: there seems to be a contradiction in the sentence. Sexual dimorphism can be observed in a turtle with a 70 cm CCL
Line 253: no references for that sentence
Lines 260-261: I agree. I would remove the previous sentence
Line 265: replace with its
Line 273: replace with quicker
Line 299: a space is needed
Line 300: 2 words attached
Fig.4: in the image, the text is not redeable.
Author Response
Reviewer 1 comments for Author:
We appreciate all your comments since they have improved the quality of our manuscript. As you explained, one of the limitations of our study was the low number of specimens. However, we have to highlight that these animals are endangered and due to this fact, it is not easy to get a large number as happens in domestic mammals. Therefore, we only access them when they appear stranding on our beaches
- Line 36: 2 words attached
We have corrected it
- 2. Line 42: the diet is not related to the environment
We have deleted the information related to its diet and added more general information
- Line 44: error
We have corrected this error and included additional information extracted from “Ernst, C.; R. Barbour, J.; Lovich. Turtles of the United States and Canada. 1994. Washington and London: Smithsonian Institution Press”.
- Line 47: no references declare this
Following your suggestion, we have redone this sentence
- Line 49: testosterone is influenced by stress
In this revised version of the paper, we have included different information related to males and females
- Line 51: in the red list they were moved to vulnerable
As you suggested, we have added that information in the introduction section
- Line 52: 2 words attached
We have corrected it
- Line 59: no scientific references allow this sentence
As you recommend, we have included specific references concerning this information
- Line 66: 2 words attached
We have corrected it
- Line 69: a space is missed
This problem has been corrected
- Line 77: with 10 animals any standardization is not scientific
Following your recommendation, we have changed the sentence and focused on measuring the eyeball and associated structures as the scleral skeleton.
- Line 82: replace with 10
We have replaced “ten” with “10”
- Line 84-85: The animals are young and subadults, the age is not so easily quantifiable
As you recommend, we have added that information.
- Lines 87 -92: I understand that healthy animals were enrolled in this study, with no signs of eye disease so it would be sufficient to say they were animals without any disease.
Following your suggestion, we have changed and redone the paragraph
- Line 86: later referred to what?
It was referred to the ophthalmologic examination. Nonetheless, we have deleted this sentence following your anterior recommendation
- Line 95: In my experience, in these animals anaesthesia is required to perform CT scans in order to prevent movement during the scans. How these animals were immobilized in the plastic container without anesthesia and without stress?
We used a specific plastic container provided by the Tafira Wildlife Rehabilitation Center, where they perform different medical procedures. Therefore, these animals are used to spend time in containers. In addition, the room temperature (ranging between 16-20ºC) produced some lethargy that could be helpful in performing the imaging study without sedation
- Line 107: in sea turtle biometric measurements CCL or SCL are used
As you recommend, we have added this abbreviation
- Line 142: take off a space
The space has been removed
- Line 143: 2 words attached
We have corrected it
- Line 160: replace with tests
As you suggested, we have replaced it with “tests”
- Line 181-182: the sentence is not clear, it needs revision
We have redone the sentence in order to clarify it
- Line 212. The sentence is incorrect, please rewrite it
Following your advice, we have corrected this paragraph
- Line 232-233 it is not correct to write “normal” loggerhead turtle. Maybe you mean healthy eyes?
As you recommend, we have corrected that sentence and added healthy eyes
- Line 244: the sentence is uncorrect: most of
We have added “most of”
- Line 246-251: there seems to be a contradiction in the sentence. Sexual dimorphism can be observed in a turtle with a 70 cm CCL
As you explain, this paragraph had many contradictions, therefore, we have redone this paragraph and deleted the information about sexual dimorphism
- Line 253: no references for that sentence
This information has been deleted
- Lines 260-261: I agree. I would remove the previous sentence
Following your suggestion, we have removed the previous sentence
- Line 265: replace with its
We have replaced “their” by “its”
- Line 273: replace with quicker
We have replaced “more quickly” by “quicker”
- Line 299: a space is needed
We have corrected it
- Line 300: 2 words attached
We have corrected it
- Fig.4: in the image, the text is not redeable.
We have redone figure 4 in order to increase the image resolution
Reviewer 2 Report
The paper is very interesting as it provides information on an endangered species. The quality of all images is very good, and also the description along the paper is complete and very detailed. Probably, one of the most relevant questions is the use of imaging techniques, which are so useful for basic research, veterinary clinic and conservation. There are just two suggestions:
1. The authors do not point out whether all procedures were approved by any ethics committee. They are using alive animals and this question is important.
2. In Table 1, the term "all eyes" shoild be replaced by "both eyes"
Author Response
Reviewer 2 comments for Author:
Firstly, we appreciate all your comments since they have improved the quality of our manuscript
- 1. The authors do not point out whether all procedures were approved by any ethics committee. They are using live animals and this question is important.
As you suggested, we have added specific information about this matter. Thus, we used the information obtained from previous CT studies to rule out the exact cause of stranding. Therefore, the rehabilitation centre requested modern methods of non-invasive diagnostic imaging to evaluate their appendicular skeleton and metabolic bone diseases and rule out foreign bodies and lesions of the internal organs. Instead of ethical approval, we got informed consent approved by the Institution (Cabildo Insular de Gran Canaria)
- In Table 1, the term "all eyes" should be replaced by "both eyes"
As you recommend, we have replaced “all eyes” by “both eyes”
Reviewer 3 Report
Morphometric study of the eyeball of the loggerhead turtle (Caretta caretta) using computed tomography (CT)is a good manuscript by Hernández et al.
The manuscript can be improved as explained below.
I recomment to use nonparametric tests (Mann Whitney U) to compare two groups due to low sample sizes.
There are also some mistakes in the manuscript for example Line 181.
There are typo and space and itailic probelmes for example -Line 36.
There are fig and Figure useages, please follow one style (Line 152,154 and 180).
The anatomy of sea turtles can also be included in the referecnes and necessary comparison can also be made accordingly.
Author Response
Reviewer 3 comments for Author:
We appreciate all your comments and suggestions since we believe that they have improved the quality of our manuscript.
- I recommend to use nonparametric tests (Mann Whitney U) to compare two groups due to low sample sizes.
Following your suggestion, we have changed the name in the article. Nonetheless, The Mann–Whitney U test is also called the Mann–Whitney–Wilcoxon, Wilcoxon rank-sum test, or Wilcoxon–Mann–Whitney test. So we have used the Mann Whitney U test.
- There are also some mistakes in the manuscript for example Line 181.
We have redone the sentence in order to clarify this statement
- There are typo and space and italic problems for example -Line 36.
Following your suggestion, we have corrected the above-mentioned mistakes
- There are fig and Figure usages, please follow one style (Line 152,154 and 180).
As you recommend, we have followed one style and used “figure” instead of “fig”
- The anatomy of sea turtles can also be included in the references and necessary comparison can also be made accordingly.
Following your recommendation, we have included this information in the manuscript
Round 2
Reviewer 1 Report
I appreciate the revision work. I disagree with point 16 of the authors' replies: claiming to keep the animals in a room with low temperatures, in my opinion, does not guarantee animal welfare.
Author Response
Answer to the reviewer’s comments:
- I appreciate the revision work. I disagree with point 16 of the authors' replies: claiming to keep the animals in a room with low temperatures, in my opinion, does not guarantee animal welfare.
- We completely agree with your concern about animal welfare. As the main responsible for the study and a member of the ethical commission of our university, we tried to avoid any disturbance in our animals. Nevertheless, the room temperature where is located the CT equipment ranges between 18-20ºC to preserve its adequate performance.